



# GCAM-Europe v7.2.0: Enhancing Policy-Relevant Climate Modelling Through Spatial and Sectoral Detail

Jon Sampedro[1,2], Russell Horowitz[1], Clàudia Rodés-Bachs[1], Dirk-Jan Van de Ven[1]

5    [1] Basque Center for Climate Change, Sede Building 1, 1st floor, Scientific Campus of the University of the Basque Country, Leioa, 48940, Spain

[2] IKERBASQUE, Basque Foundation for Science, Plaza Euskadi 5, 48009 Bilbao, Spain.

*Correspondence to*:    Jon    Sampedro    (jon.sampedro@bc3research.org)    and    Dirk-Jan    Van    de    Ven (dj.vandeven@bc3research.org)

10   **Abstract.**

Integrated Assessment Models (IAMs) serve as critical instruments for scenario-based analysis and have been instrumental in informing environmental policy at both global and regional scales. However, their limited geographical and sectoral scope constrains their ability to evaluate comprehensive policy packages such as the European Union's Fit-for-55. GCAM-Europe, an expansion of the well-stablished Global Change Analysis Model (GCAM) addresses this gap by explicitly representing

15   energy, land use, agriculture, water, and emissions systems for European Member States and key non-EU countries. Operating within a global framework, the model enables integrated assessment of policy impacts both across and within European regions, while also capturing spillover effects in other regions in the world.  As an open-access and continuously evolving platform, it provides a valuable tool for European researchers, policymakers, and stakeholders to design, test, and evaluate climate and environmental strategies that support a just and effective climate transition.



## 1. Introduction

Integrated assessment models (IAMs) are "*simplified representations of complex physical and social systems, focusing on the interaction between economy, society and the environment*" (IAMC, 2022). Over the past four decades, IAMs have played a relevant role in shaping environmental policy and advancing research at both global and regional levels (Weyant, 2017). In recent years, the number of IAM-related publications addressing climate change has grown exponentially, and IAMs have become integral to the development of recent IPCC reports (IPCC, 2022), supporting scenario analysis, mitigation strategies, and long-term climate projections (van Beek et al., 2020; Fisher-Vanden and Weyant, 2020). The Global Change Analysis Model (GCAM) is a well-reputed multisector IAM developed and maintained at the Pacific Northwest National Laboratory's Joint Global Change Research Institute (Calvin et al., 2019; JGCRI, 2025). The model is designed to explore the interdependencies between the energy, agriculture, forestry and land use (AFOLU), water, and climate systems in 32 geopolitical regions across the globe. It enables the analysis of alternative "what-if" type scenarios through a unified computational platform, with projections extending up to the year 2100. GCAM is a fully open-access community model that has been used in major cross-regional analyses, including the IPCC reports (IPCC, 2022), the development of the Shared Socioeconomic Pathways (Calvin et al., 2017), or the recent Scenario Model Intercomparison Project for CMIP7 (van Vuuren et al., 2025).

Nevertheless, GCAM and other existing IAMs are limited to assess comprehensive packages like the EU's Fit-for-55 or the Inflation Reduction Act (IRA). Global models often lack the geographic and sectoral detail needed to capture national and regional differences, while national models typically miss cross-border interactions and wider economic dynamics (Anderson and Jewell, 2019; Brutschin et al., 2021; Keppo et al., 2021). IAMs generally focus on identifying "least-cost" pathways to emission reduction targets (van de Ven et al., 2023), which involves applying a uniform carbon price across all regions and sectors, with little attention to feasibility (Bertram et al., 2024) or non-emission outcomes (Geels et al., 2016). As a result, these models often produce scenarios that diverge significantly from political realities, such as over-reliance on negative emissions technologies (NETs) (Lamb et al., 2024) and on land removals (Dooley et al., 2024), or unrealistic policy assumptions (Gambhir et al., 2019). In addition, IAMs often fail to adequately represent the diversity of household behaviour and consumption, which critically limits their ability to assess inequality impacts and the broader social and distributional consequences of climate transitions, which are essential to ensure no one is left behind in the shift to a low-carbon future (Emmerling et al., 2024; Low et al., 2025). This underscores the critical need to develop new, and refine existing, IAMs (Braunreiter et al., 2021; Koasidis et al., 2023; Skea et al., 2021). Particularly, enhancing the geographical resolution of IAMs is essential to accurately represent the heterogeneity of socio-economic systems, institutional capacities, and political contexts at the national level. Such granularity enables more detailed, bottom-up climate policy modelling beyond stylized assumptions like uniform carbon pricing, thereby improving the feasibility, relevance, and robustness of projected outcomes.





The GCAM community has already several regional versions of the models for different regions over the world that include the USA (GCAM-USA; (Binsted et al., 2022), China (GCAM-China; CGS, 2025), Korea (GCAM-Korea; Jeon et al., 2021), Middle East (GCAM-KSA), South America (GCAM-LAC), Canada (GCAM-Canada) or India (GCAM-India). These regional versions are better suited for modelling bottom-up environmental policies and assessing their system-wide implications at both

national and subnational scales. Therefore, they have garnered significant interest from a diverse array of stakeholders. For instance, GCAM-USA has been widely employed to evaluate the impacts of existing climate policies at the national and state levels (Bistline et al., 2025; Zhao et al., 2024), with its outputs informing both academic research and decision-making in policy and industry. Similarly, the recent launch event for the latest version of GCAM-China drew considerable attention, with nearly 200 participants attending in person and over 10,000 viewers joining online. Within this framework, we are confident

that GCAM-Europe will serve as a powerful tool for in-depth analysis of European climate policy and will receive strong interest from key stakeholders involved in the climate transition. In addition, unlike existing regional versions of GCAM, GCAM-Europe explicitly represents land use, agriculture, water resources, and energy extraction at the subregional level. This constitutes a significant contribution to the European IAM community, enabling more precise assessments of resource dynamics and policy impacts across diverse European contexts.

**2.   Model description**

GCAM-Europe's geographical disaggregation is highly-detailed for the European continent. While core GCAM divides the European continent into five different regions, namely EU-12, EU-15, Europe Eastern, Europe-non-EU, and European Free Trade Association (EFTA), in GCAM-Europe all European countries are disaggregated into individual model regions (Figure 1) for the energy-economy system. These include the 27 Member States of the European Union (EU) and additional key non-

EU countries (e.g., Switzerland or the United Kingdom). In terms of land-use and water systems, GCAM-Europe disaggregates 119 land regions, which result from the interactions between geopolitical regions (countries) with the hydrologic basins. Having this level of detail enables us to explore the country-level effects of European policy packages or transformational strategies, as well as the potential international effects (e.g., carbon leakages).



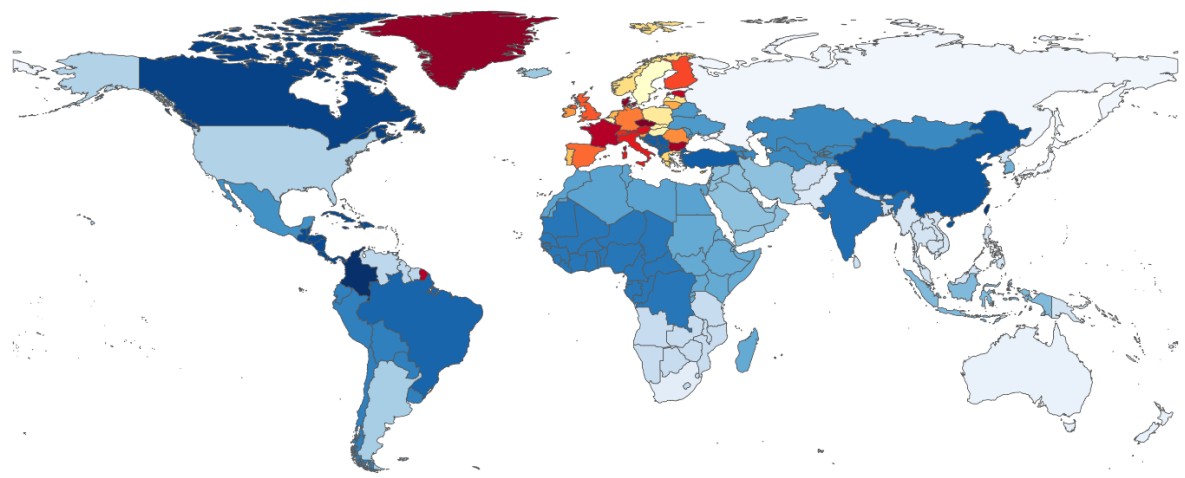

**Figure 1: Regional disaggregation in GCAM-Europe.** Yellow-to-red colours represent the regional groups that are explicitly disaggregated for this model. White-to-blue regions represent model regional groups that were part of the original model.

GCAM-Europe replaces the default (international) data sources for all newly defined European countries with Europe-specific data, such as energy statistics from Eurostat (Eurostat, 2025a), whenever available. If certain countries lack coverage in these European datasets, alternative sources are used, most commonly reverting to the default GCAM data, such as IEA energy statistics (IEA 2022). Population projections are also replaced by specific data from the EU Ageing Report 2024 (European Commission, 2024), ensuring consistency with official demographic and economic outlooks.

Beyond the regional disaggregation and data replacement, GCAM-Europe includes additional features compared to the core version, including the enhanced representation of electricity grids and regional trade, as well as demand segments, which allows to have more sectoral detail in the power sector. In final energy demand, predominantly in building energy demand, new demand categories are included as well as new technologies like heat pumps, driven by high data availability for European countries.

## 2.1 Power system

The electricity system in GCAM-Europe is modelled as an interconnected system structured around grid regions, load segments, and inter-segment storage. Like in other regional GCAM versions (e.g., GCAM-USA), the electricity system in GCAM-Europe is subdivided into several "grid-regions", in which all electricity supply and demand can flow unrestricted, while electricity trade between grid-regions is relatively more constrained. Figure 2 shows the grid region structure for GCAM-Europe, which we have designed largely following the seven "wholesale regions" as defined in the quarterly European electricity market reports, and an additional region joining together the electricity grid of Moldova and Ukraine. This leaves 3





remaining European countries without being integrated in larger grid regions: Belarus, Iceland and Turkey. The national grids of Belarus and Iceland will act as standalone grids without interconnection, as real-world interconnection with other European countries is insignificant. For Turkey, grid interconnection with the rest of Europe is represented.

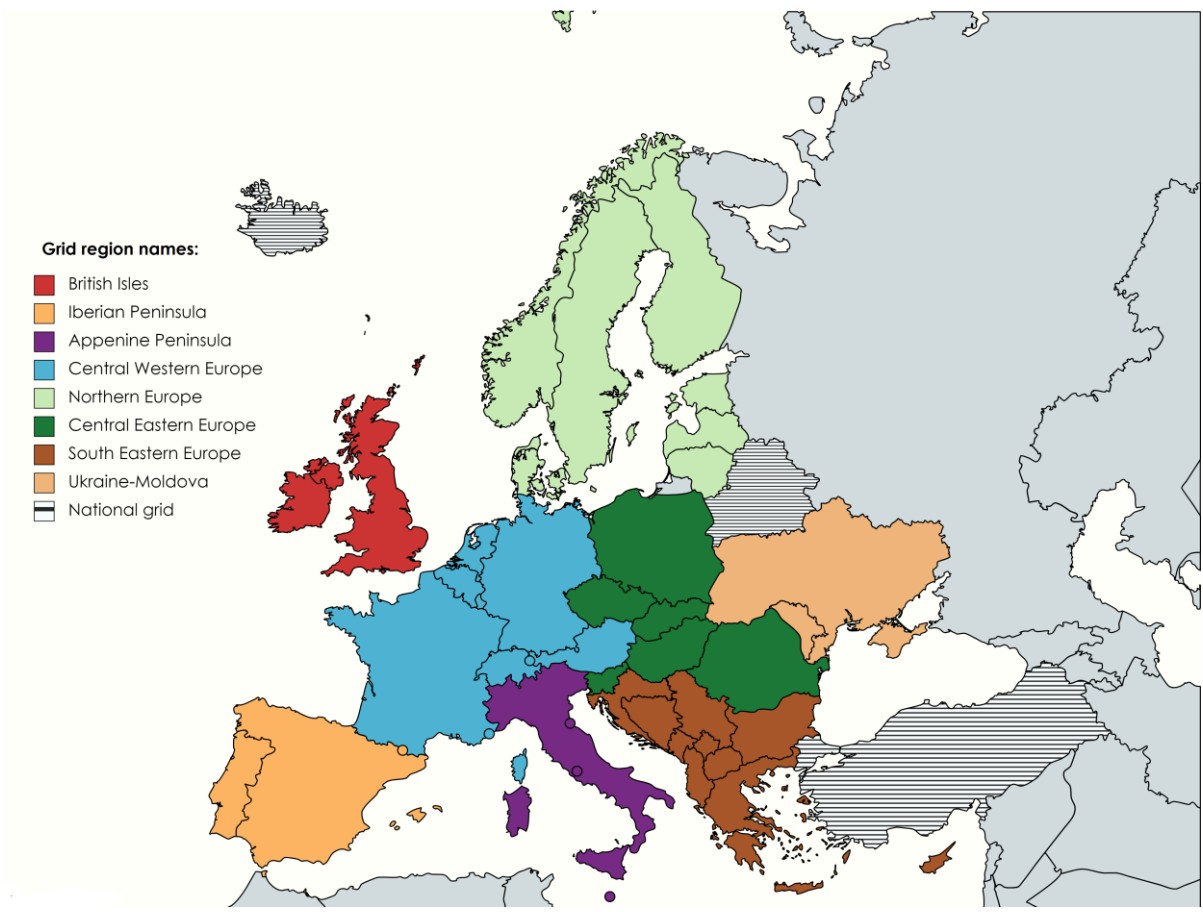

**Figure 2: Electricity grid regions GCAM-Europe**


The representation of the electricity system is based on the "National trends" scenario of the "Ten-Year Network Development Plans" (TYNDP) 2024 process by the European Network of Transmission System Operators for Electricity (ENTSO-E). This scenario serves as a reference for grid investment planning and electricity demand projections across Europe (ENTSOE, 2024). Load segmentation is derived using data from the openTEPES electricity model (Ramos et al., 2022), resulting in four distinct load segments: base load, intermediate, subpeak, and peak. This segmentation captures temporal variations in electricity demand and supply with greater accuracy and supports detailed analysis of storage technologies and their role in balancing the grid across time periods and regions. Each grid region and load segment operates its own electricity market, with prices




typically lowest during off-peak periods and highest during peak demand. These price dynamics can incentivize investment in storage technologies, while regional price differences may encourage cross-border electricity trade.

## 2.2. Commodity trade structure

GCAM-Europe incorporates a trade structure that reflects the European Economic Area (EEA) and the internal free-trade
market among EU member states, Norway, and Iceland (Figure 3). By default, commodity trade in GCAM, including energy, food, and selected industrial commodities such as steel and ammonia, is governed by Armington trade specifications (Armington, 1969), which define the substitutability between domestically produced and imported goods, both at the international level and within individual regions. In GCAM-Europe, EEA countries are treated as a single trading block in relation to other GCAM regions in the international market, including the newly separated United Kingdom. Within the EEA,
intra-regional trade is modelled with an additional layer of Armington specifications to capture trade flows among member countries. While international trade costs between EEA and non-EEA regions remain the same as in the default GCAM framework, trade within the EEA is assigned lower costs, reflecting reduced trade barriers within the internal markets. Beyond improved trade realism, this enhanced structure facilitates the representation of shared trade policies across the EU and EEA, such as the Carbon Border Adjustment Mechanism (CBAM), enabling more nuanced and policy-relevant modelling in future
analyses.

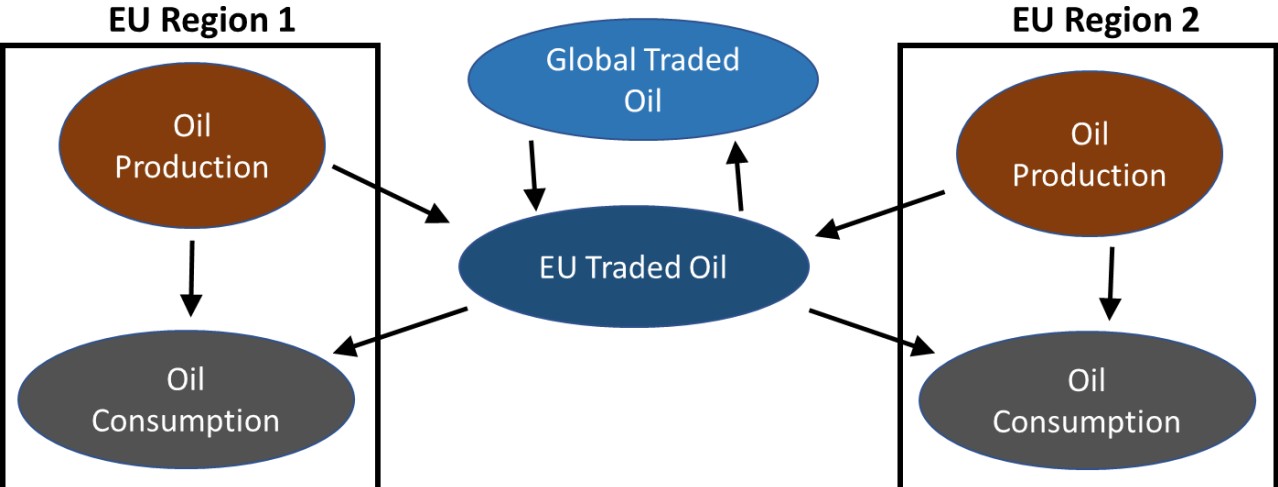

**Figure 3: Updated trade structure in GCAM-Europe**



## 2.3 Sectoral and technological coverage in the buildings sector

Leveraging the richer and more detailed data available for European countries, the model has been expanded across multiple dimensions with a particular emphasis on building energy demand. The core GCAM version disaggregates residential energy consumption into cooling, heating and non-thermal (e.g., appliances) services. In GCAM-Europe, following the categorization in Eurostat (Eurostat, 2025b), we introduce new demand categories, namely hot water, cooking, and various household appliances, enabling more accurate and granular modelling of household energy consumption patterns.

In addition, GCAM-Europe includes a detailed representation of heat pump technologies, which are not explicitly modelled in the core version of GCAM. The model distinguishes between several types of systems: air-source heat pumps that extract energy from the outside air ("air-air"), water-source heat pumps that extracts heat from the outside air and transfers it to a water-based heating system ("air-water"), and ground-source heat pumps that draw energy from the ground ("geo-water"). GCAM-Europe also incorporates solar thermal technologies for both space and water heating, providing a more comprehensive view of low-carbon heating solutions.

In terms of household structure, GCAM-Europe includes consumer heterogeneity in the residential energy sector in the form of income deciles, which is aligned with the current structure in the core version of the model. As a next step, we plan to enhance this representation by integrating empirical data from national Household Budget Surveys (HBS) (Eurostat, 2025c), which are available for all EU member states. This improvement will allow for a more realistic calibration of household energy consumption and expenditure patterns, thereby increasing the accuracy and policy relevance of model outputs (see *Discussion and Conclusion*).





# 3. Results

## 3.1. Socioeconomics

The socioeconomic projections in GCAM-Europe are aligned with the moderate demographic and economic growth defined in the SSP2 narrative (O'Neill et al., 2014), which is the same assumption in the baseline scenario of the core version of the model. However, the population trajectories for different countries in GCAM-Europe have been updated using EU-specific information from the EU Ageing Report 2024 (European Commission, 2024). This results in higher population projections at European level, compared to the core. Figure 4 shows that European population peaks and start declining at the middle of the century. However, in the core model, population accounts for 685M and 632M people in 2050 and 2100, respectively, while these values represent 696M and 649M in the GCAM-Europe baseline. In terms of GDP, GCAM-Europe does not include any update and the projections are completely aligned with the moderate economic growth in the core model (Fig S1).

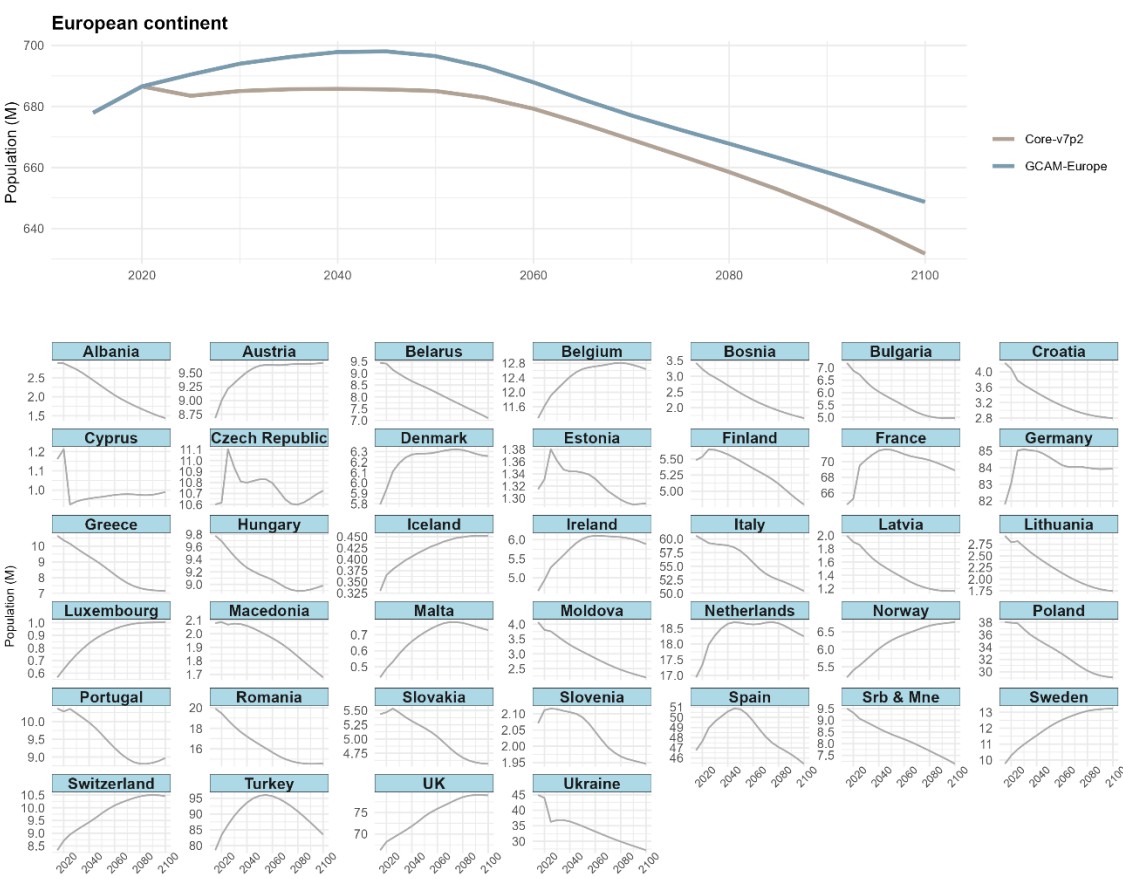

**Figure 4: Population by period, region, and scenario (Million).** The upper panel compares population projections between the GCAM-Europe and the core baseline scenarios. The lower panel shows population by European country in the GCAM-Europe baseline.



## 3.2. Energy

In the GCAM-Europe baseline scenario, energy consumption gradually increases over time, peaking in the last quarter of the century, in line with socioeconomic trends and influenced by the adoption of more efficient technologies across end-use sectors. Primary energy consumption in Europe increases from 76 EJ in 2015 to 90 EJ in 2070, and then slightly declines to 87 EJ by the end of the century. Total final energy consumption shows a similar trend, increasing from 65 EJ in 2015 to 69 EJ in 2070, and then decreasing to 67 EJ by 2100. These trends in both primary and final energy consumption are consistent with

the projections in the core version of the model. However, the implementation of new features, such as the grid regions and load segments in the power system, as well as the highly-efficient heat pump technologies implies that the values in the GCAM-Europe baseline are lower than in the core GCAM model (Figure 5). In the core version, primary energy in Europe increases to 100 EJ in 2070 and declines to 97 EJ in 2100. Likewise, final energy achieves around 77 EJ and 75 EJ in 2050 and 2100 respectively, which are higher than the values in the GCAM-Europe baseline.

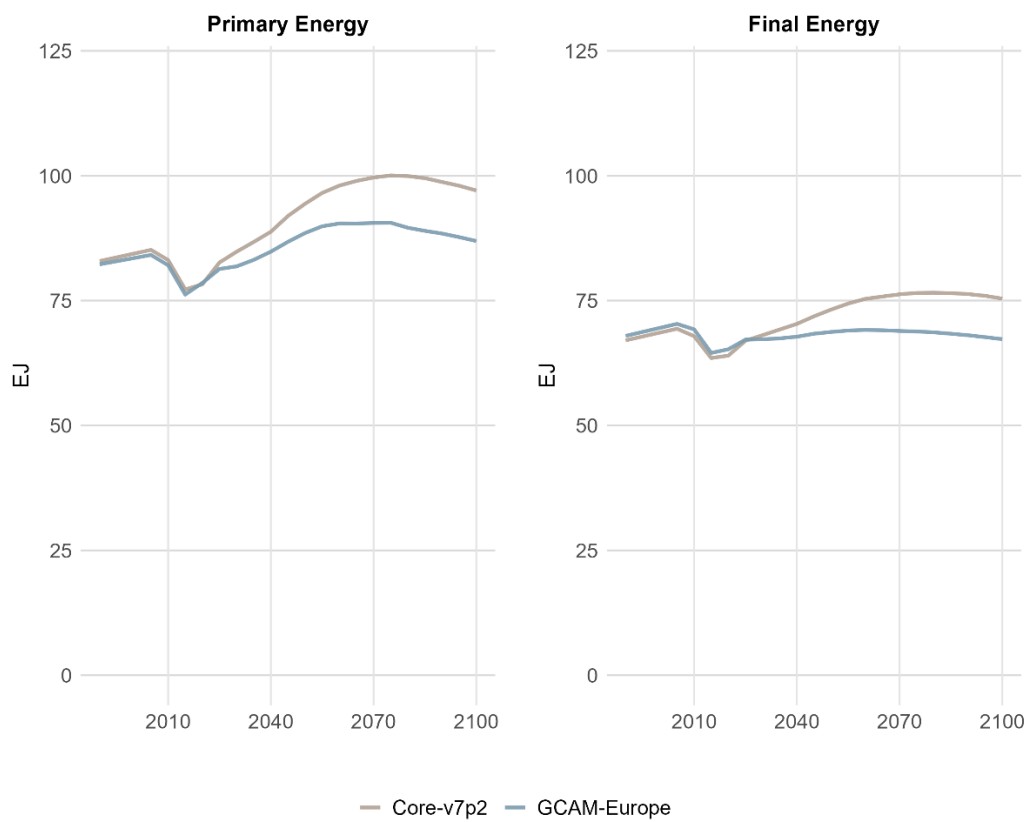


**Figure 5: Primary energy (left) and final energy (right) in Europe by period and scenario (EJ).**





Fossil fuels are the main source of primary energy during the entire time horizon in the GCAM-Europe baseline scenario.
However, the share of renewable energy gradually increases as technological advancements reduce costs and therefore increase

their competitiveness. In 2015, fossil fuels represent around 80% of total primary energy at European level and by 2050 this
share is reduced to around 70%. This share is similar to the value observed in the core GCAM version. However, there are
some differences across fuels. In GCAM-Europe there is a higher penetration of biomass technologies, reaching about 13 EJ
in 2050 and comprising 15% of the primary energy mix. This value is smaller in GCAM core, as in the same period biomass
consumption accounts for 11 EJ (12% of the mix). Contrarily, the baseline scenario in GCAM-Europe shows a more

pessimistic penetration of wind energy. In 2050, wind power achieves 1.8 EJ, representing around 2% of the primary energy
mix. In the core model, these values rise to 4.5 EJ and 4%, respectively. A comparison of primary energy across scenarios in
2050 is presented in Figure 6.

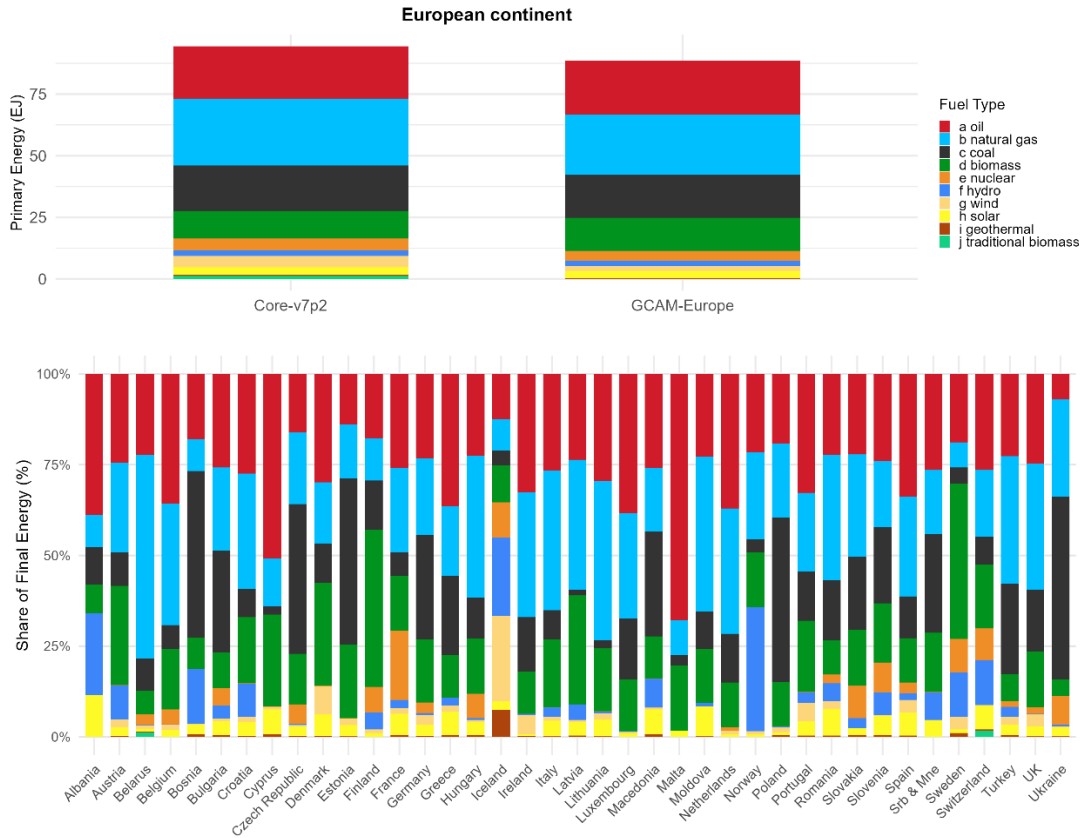

**Figure 6: Primary energy consumption in 2050 by scenario, region, and fuel (EJ).** The upper panel shows total primary energy for the
European continent (EJ). The lower panel shows the share of different fuels in each country's primary energy mix (%) in the GCAM-
Europe baseline. Total primary energy by European country is shown in the Supplementary Information (SI) Fig S2.





Total final energy shows similar trends and differences across scenarios and fuels (Fig S3). Focusing on final energy by sector,

industry is projected to be the most energy-intensive sector during the entire time horizon. Industrial energy demand increases from 24 EJ in 2015 to 30 EJ by 2070, driven by growing industrial activity associated with the projected population growth and economic expansion. Then, it slightly declines by the end of the century to 28 EJ. Energy consumption for the transportation sector shows a moderate and steady increase throughout the entire time horizon, with the associated energy consumption rising from 19 EJ in 2015 to 22 EJ by 2100. This rise is primarily driven by growing demand for freight

transportation services. On the contrary, the building sector presents a decrease over time in energy consumption from 21 EJ in 2015 to 16 EJ in 2100. This is largely due to most European households approaching energy satiation levels, meaning that rising incomes no longer lead to increased demand for thermal services such as heating and cooling. As residential demand for energy services stabilizes, the growing adoption of efficient heat pump technologies (Fig S4), which become increasingly competitive in the near future, directly contributes to the overall decline in energy use within the sector. There are some

differences when comparing these trends with the GCAM core baseline scenario. Energy demand in the transportation sector is slightly higher in the GCAM-Europe baseline, whereas demand in both the industrial and building sectors is higher in the core scenario. The disparity is especially pronounced in the building sector, considering that the core version does not explicitly model the highly efficient heat pump technologies. This information for a representative year (2050) is summarized in Figure 7.







**Figure 7: Final energy consumption in 2050 by scenario region, and fuel (EJ).** The upper panel shows total final energy for the European continent (EJ). The lower panel shows the weight of different sectors in each country's final energy consumption (%) in the GCAM-Europe baseline. Total final energy by European country is shown in the Supplementary Information (SI) Fig S5.

Focusing on the power sector, electricity production in the GCAM-Europe baseline scenario shows an steady increase during the entire century, from around 14 EJ generated in 2015 to 24 EJ in 2100. However, this growth is smaller than the one in the core baseline, which exceeds the 29 EJ of electricity generation by 2100 (Fig S6). The notable differences between the core and GCAM-Europe are driven by variations in total final energy demand, as well as differences in the representation of the power system.

The share of fossil fuel in total electricity generation is almost similar in the two model versions. We see that in the two scenarios coal and gas still represent around 20% and 15% of the European electricity mix by 2050. However, notable





differences emerge between GCAM-Europe and the core model version in the deployment of renewable energy sources within the electricity sector, largely driven by the load segments, inter-segment storage, and grid regions included in GCAM-Europe. For example, grid storage represents around 6% of the electricity mix by 2050 in the GCAM-Europe baseline, which is not

represented in the core. Likewise, there are large differences in wind power generation. While it represents around 18% of the total electricity mix in 2050 in the core, this share decreases to 7% in the GCAM-Europe baseline. The share of total solar power is also similar across scenarios, but there are some technological differences. In GCAM-Europe there is a higher penetration of distributed rooftop photovoltaics, while centralised photovoltaics power sources are the main solar energy source in the core. Nuclear energy is also relatively similar in the two scenarios, accounting for around 18% of the European electricity

mix by 2050.

There are also large differences across European regions, with different countries showing completely different electricity mixes, depending on their access to alternative energy sources (Figure 8). By 2050, some regions are projected to rely almost entirely on renewable energy (e.g., Switzerland). In contrast, fossil fuels are expected to continue playing a major role in other countries, particularly in Eastern Europe (e.g., Bosnia). We also observe notable differences across electricity system segments,

with intermittent technologies playing an increasingly important role in baseload generation, while fossil fuels remain dominant during peak and sub-peak demand periods.



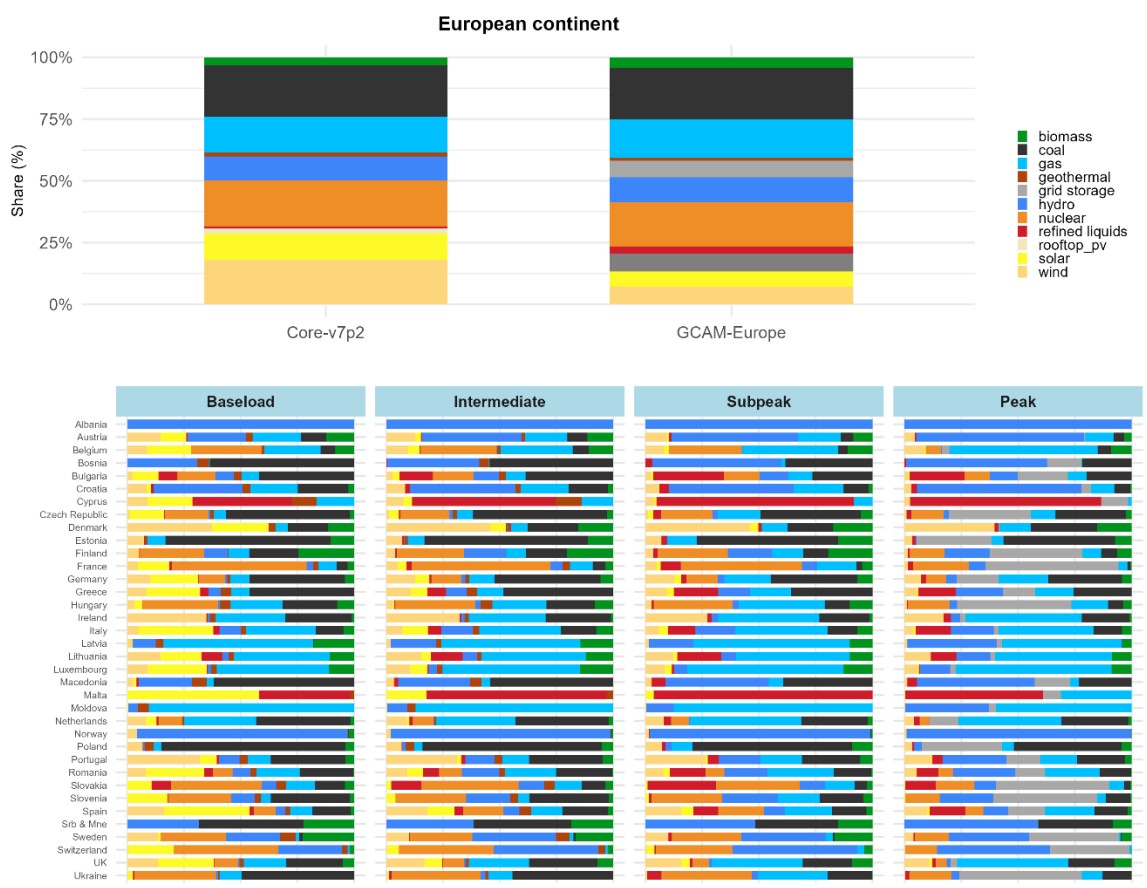

**Figure 8: Electricity generation in 2050 by scenario region, and source (EJ).** The upper panel the total electricity mix for the European continent (%). The lower panel shows the weight of different sources in each country's final electricity mix (%) in the GCAM-Europe baseline. Total electricity generation by European country is shown in the Supplementary Information (SI) Fig S7. Belarus, Iceland and Turkey are not included as they are not connected to the European grid regions.






### 3.3. Land and water

The higher disaggregation and new features incorporated into GCAM-Europe have some direct and indirect impacts on the
distribution of land compared to the core version of the model (Figure 9). In GCAM, there are some land types that are
exogenously defined and do not compete with other land types. These include the urban, tundra and "rock and dessert"
categories, which are therefore identical in the two scenarios. Some natural land types, such as shrubs or grass, also show
minor variations between GCAM-Europe and the core. The most significant differences are found in the distribution of
pasturelands and forests. GCAM-Europe projects a larger share of grazed pastures, reaching up to 490 thousand km2 by the
end of the century, which is substantially higher than the 344 thousand km2 projected by the core model. In contrast, the area
of non-grazed ("other") pastures is notably larger in the core version of the model, especially in the near term, with projections
of 611 thousand km2 in 2030 compared to 499 thousand km2 in GCAM-Europe. A similar dynamic is observed in the
distribution of forestland. The core model has more land allocated for managed forest, representing up to 1720 thousand km2
in 2050, higher than the 1420 thousand km2 in GCAM-Europe. However, the share of natural (unmanaged) forest is larger in
the GCAM-Europe baseline. Land distribution varies significantly across European countries. While cropland and forests
represent the majority of land use in many regions, pasture is particularly prominent in some areas, such as the United
Kingdom. In Scandinavian countries, such as Denmark, tundra and "rock and desert" areas also account for a notable portion
of the total land area (Figure 9).



**Figure 9: Land allocation by scenario, region, period and type.** The upper panel shows the total land area for the European continent by scenario, period and land type. The lower panel shows the weight of different land types in each country for different time periods (%) in the GCAM-Europe baseline.

Total water withdrawals for the European continent in GCAM-Europe show a moderate and steady increase over time, rising from 330 km³ in 2020 to 380 km³ in 2085, followed by a slight decline to 365 km³ by 2100 (Figure 10). The figure also shows how the projected differences in energy mix and land allocation between GCAM-Europe and the core model also result in changes to water systems, with GCAM-Europe showing a slightly higher water withdrawal volume by the end of the century. Water withdrawal levels vary across European countries, with some countries such as Germany, France, and Turkey recording notably higher volumes. While certain countries, like France, exhibit an upward trend in water withdrawals, the overall growth across most of Europe remains relatively moderate. In fact, some countries, such as Poland, show a decline in their water withdrawal volumes over time.



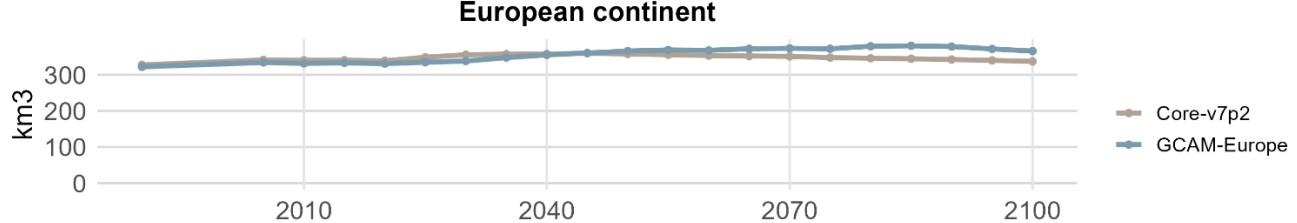

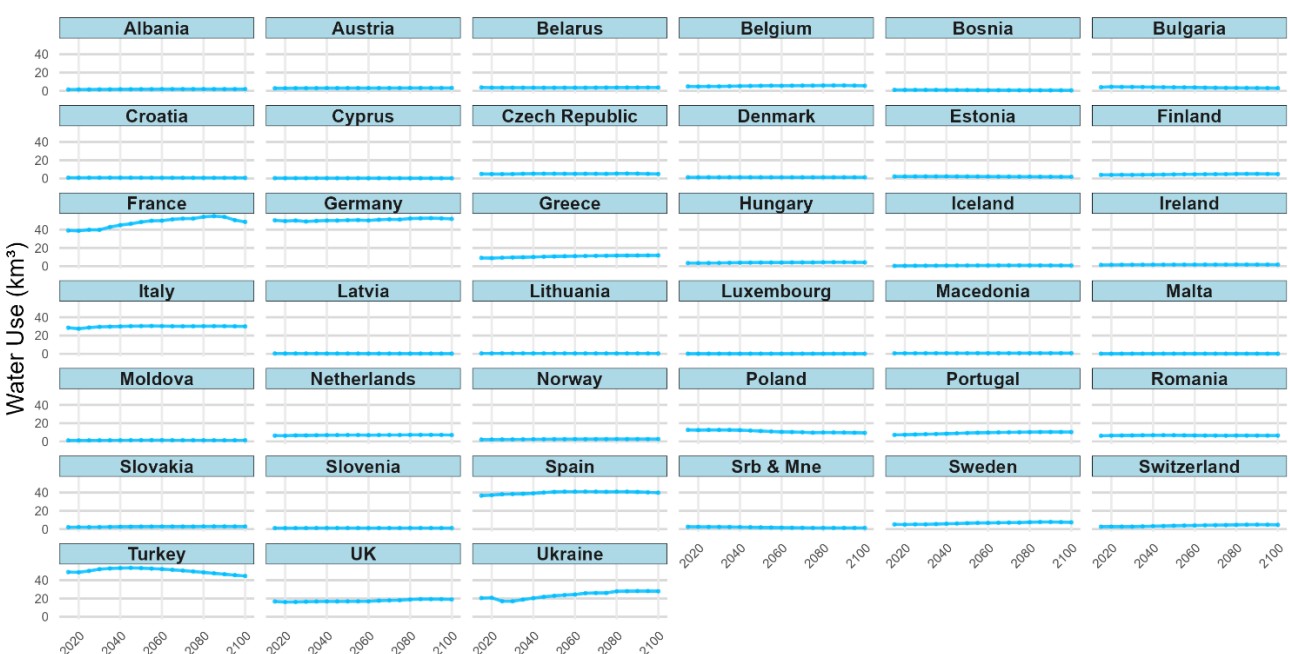

**Figure 10 Water withdrawals by scenario, region, and period (km3).** The upper panel shows total water withdrawals for the European continent by scenario and period. The lower panel shows water withdrawals by period and country in the GCAM-Europe baseline.






### 3.4. Emissions

Future emission projections of greenhouse gases and air pollutants in GCAM-Europe substantially varies across regions and species (Figure 11). Total carbon dioxide (CO2) emissions remain relatively constant at European level, slightly rising from 1178 MTC in 2015 to 1220 MTC in 2100. Detailed CO2 emission projections for different European countries are provided in the SI (Fig S8). Similar trends can be observed in the emissions of other greenhouse gases such as methane (CH4) and nitrogen dioxide (N2O).  In contrast, emissions of air pollutants generally decrease over time, with the exception of ammonia

(NH3), which increases due to rising livestock production. These decreasing trends in air pollutants are driven by assumed reductions in emission factors (EFs) over time resulting from technological advancements and stricter air quality regulations. There are also some differences across GCAM-Europe and the core model, particularly in terms CH4 and sulphur dioxide (SO2) emissions. The divergence in CH4 emissions are associated with changes in livestock. For example, methane emissions attributable to the production of beef account for 6.5 Tg and 5 Tg in the core and GCAM-Europe baselines, respectively in

2100. The differences in SO2 emissions are mainly driven by the lower final energy consumption in the GCAM-Europe baseline.

Figure 11 also highlights significant differences in GHG and air pollutant emissions across European countries, with certain nations emerging as major contributors for specific species. Germany leads emissions of CO2 with nearly 192.5 MTC in 2050, followed closely by Turkey (146.7 MTC), the UK (131.7 MTC), and France (86.3 MTC), reflecting their larger energy

demands. CH4 emissions are highest in Turkey (3.96 Tg in 2050), Ukraine (3.51 Tg), and France (3.01 Tg), which is directly related to the size of their of agricultural and waste sectors. In the case of NH3, primarily linked to livestock and fertilizer use, Turkey, Germany, and France show the largest emissions with values in 2050 accounting for above 0.7 Tg. For nitrous oxide (N2O), emissions are more evenly spread, but Turkey, France, and Germany remain the top contributors. Nitrogen oxides (NOx), mainly from combustion processes, show the highest emissions in Turkey, Germany, and Spain, each exceeding 0.5 Tg

in 2050.  Emissions of primary particulate matter (PM2.5), namely black carbon (BC) and organic carbon (OC), have their highest emissions in Germany, Poland, and Turkey (BC), and Belarus, Italy, and Ukraine (OC). Finally, sulphur dioxide (SO2) emissions, linked to fossil fuel combustion, are highest in Turkey, followed by far by Ukraine, and Poland, highlighting continued reliance on sulphur-intensive energy sources in these regions.







**Figure 11: GHG and air pollutant emissions by scenario, region, period and specie.** The upper panel shows total emissions for the European continent by scenario, period and specie (MTC and Tg). The lower panel shows emissions by period, country and specie for 2050 in the GCAM-Europe baseline.





### 3.5. Trade dynamics

The implementation of the double-Armington structure to represent trade dynamics in GCAM-Europe change the relation between domestic versus imported commodities in the different European countries (Figure 12). Taking 2050 as a reference

year, we observed several differences across trade commodities. For crude oil, both scenarios project heavy reliance on imports, but in GCAM-Europe this dependence is slightly increased (20.22 EJ imported versus 18.62 EJ in the core). In the case of solid fuels such as coal and biomass, domestic production continues to exceed imports in both scenarios; however, the structural shift in GCAM-Europe results in increased reliance on imports and a corresponding decline in domestic output. By 2050, imports of biomass and coal are projected to rise to 4.63 EJ and 8.61 EJ, respectively, under GCAM-Europe, both

noticeably higher than the 2.53 EJ and 7.20 EJ observed in the core scenario. Natural gas shows a decline in both domestic production and imports in GCAM-Europe compared to the core, driven by the reduced demand for gas-based energy technologies projected in this scenario. In the case of crops, the model assumes a "single European market" for agricultural commodity trade, which results in only minor differences between scenarios in the balance between domestic production and imports. GCAM-Europe projects a modest increase in domestic crop production (1,450 Mt compared to 1,395 Mt in the core)

alongside a slight reduction in imports (398 Mt versus 418 Mt) by 2050. The double Armington structure implemented in GCAM-Europe has a pronounced impact on the import patterns of several key commodities, including iron and steel, ammonia, and livestock. In the GCAM-Europe baseline for 2050, imports of these commodities rise significantly, reaching 129 Mt for iron and steel, 12 Mt for ammonia, and 45 Mt for livestock, substantially higher than the corresponding values in the core scenario, which account for 69 Mt, 6 Mt, and 13 Mt, respectively. In contrast, domestic production of these same commodities

is lower in the GCAM-Europe baseline: iron and steel production decreases from 116 Mt in the core to 80 Mt, ammonia declines from 18 Mt to 16 Mt, and livestock production decreases from 273 Mt to 244 Mt.





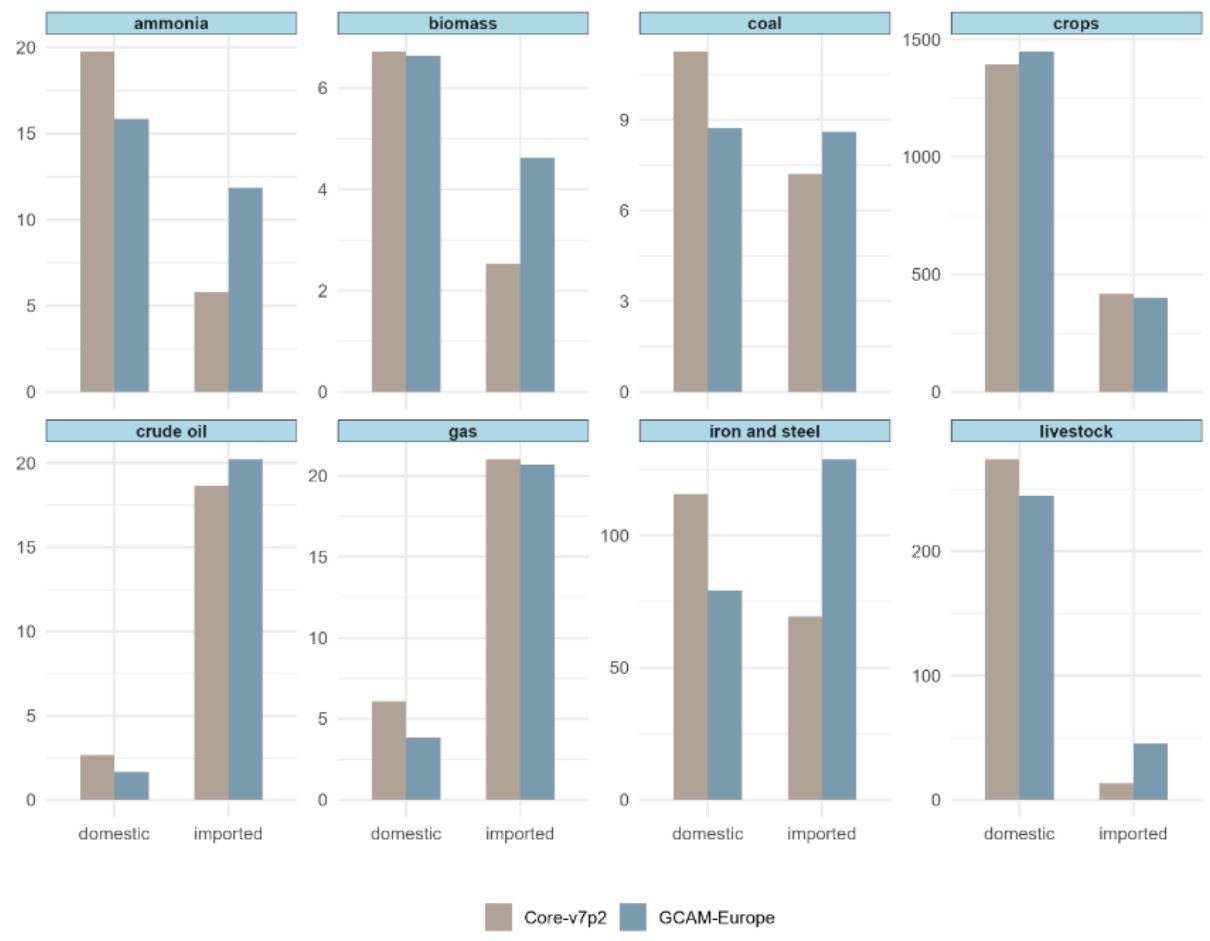

Figure 12: Domestic production and imports in 2050 by scenario and commodity (Mt and EJ).






## 4.  Implementation of a climate policy

The aim of this section is to see how GCAM-Europe responds to the implementation of an ambitious climate policy and analyse the main resulting changes in terms of energy system and emission pathways. The "policy" scenario includes a CO2 emission constraint for EU-27 compatible with the Nationally Determined Contribution (NDC) in 2030 and the 2050 Net-Zero target (van de Ven et al., 2023). It also introduces a carbon tax for other European and global regions, consistent with their respective NDCs and long-term climate goals.

The policy has a direct impact on primary energy consumption at European level, affecting both total energy demand and the composition of the energy mix (Figure 13). At EU level, there is a rapid and pronounced decrease of fossil fuel demand, which produces several co-benefits in terms of air quality or energy security. In the policy scenario, coal, gas, and oil consumption decrease by 7.4, 6.6, and 6.4 EJ, respectively, in 2050, and by 8.5, 7.3, and 9.4 EJ in 2100, compared to the baseline. The reduced fossil fuels demand primarily offset by biomass, nuclear and other renewable energy sources, such as wind or solar. At the EU level, biomass use rises by 13 EJ in 2050 and 17 EJ in 2100 in the policy scenario compared to the baseline. Nuclear energy also sees an increase of approximately 2.6 EJ in 2050 and 3.8 EJ in 2100. Solar and wind energy together contribute an additional 2.2 EJ in 2050 and 2.4 EJ in 2100 relative to the baseline. The non-EU European regions show similar trends, with biomass and other renewable energy sources consistently replacing fossil fuel demand throughout the entire century in the policy scenario. While all European countries show a consistent decline in fossil fuel demand, the choice of (non-emitting) replacement technologies varies by country (Figure 13). This variation is influenced by country-specific factors such as current and projected costs of resources and technologies, as well as historical preferences for certain energy carriers. For example, nuclear energy is expected to play a major role in France due to its existing nuclear infrastructure. In contrast, countries like the UK are projected to see a large share of wind power in their primary energy mix.





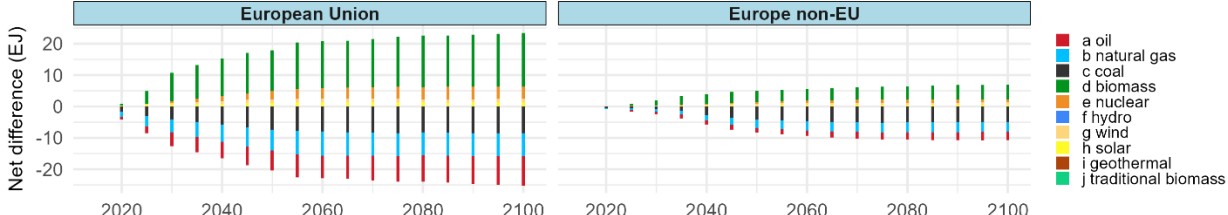

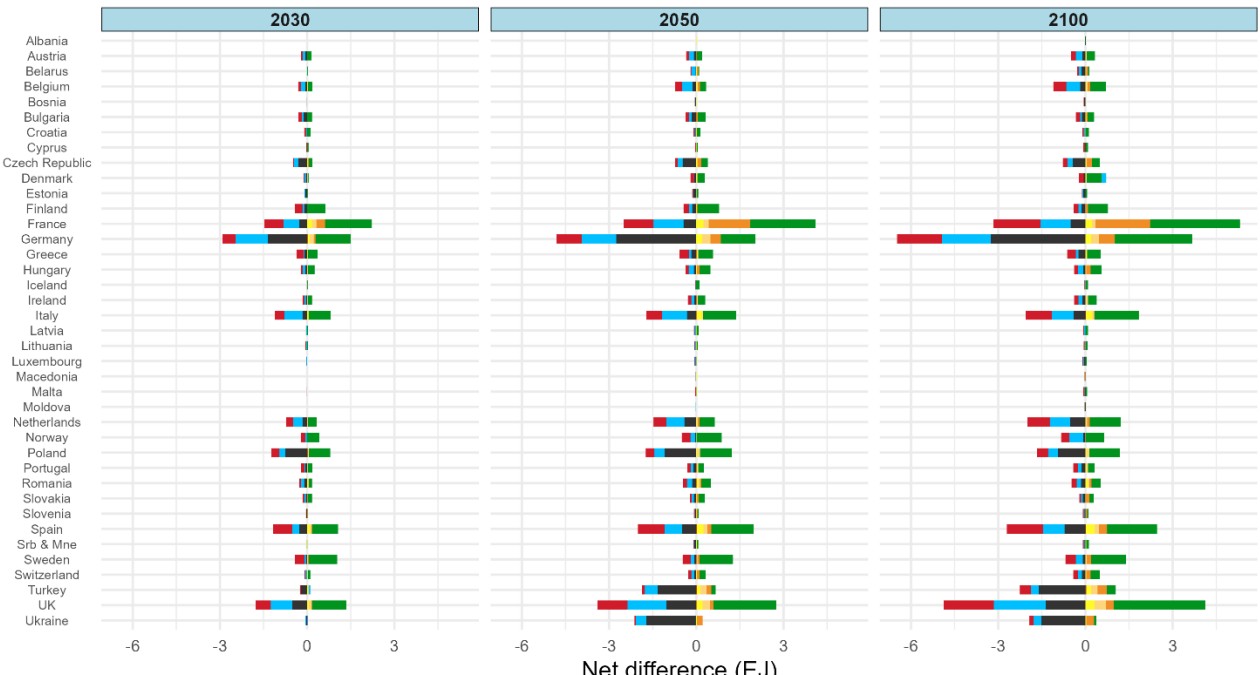

**Figure 13: Difference in primary energy demand between the policy and the baseline scenarios by period, region, and fuel.** The upper panel shows absolute differences (*Policy - Baseline*) in primary energy for the entire EU and non-EU regions by period, region and fuel (EJ). The lower panel shows the country level differences by period and fuel (EJ).

The power sector shows similar trends, with non-emitting technologies replacing fossil fuels in the electricity mix under the policy scenario, both in the EU and other European regions. (Figure 14). However, in this sector, nuclear energy is projected to be the main substitute, with an increase of 2.5 EJ in 2050 and 3.7 EJ in 2100 under the policy scenario compared to the baseline. This significant growth is primarily driven by expected nuclear deployment in Germany and France. Other countries, such as Spain and the UK, are projected to see a larger increase in solar and wind power, which become the primary substitutes for reduced fossil fuel use. Notably, several countries show some deployment of carbon capture and storage (CCS) technologies, particularly in combination with biomass (i.e., BECCS). However, the projected contribution of CCS to the



electricity mix remains limited and does not constitute a significant share in any European country. Finally, Figure 14 illustrates
an overall increase in total electricity consumption at both the EU and non-EU levels, with particularly notable growth in
certain countries, such as France or Spain.

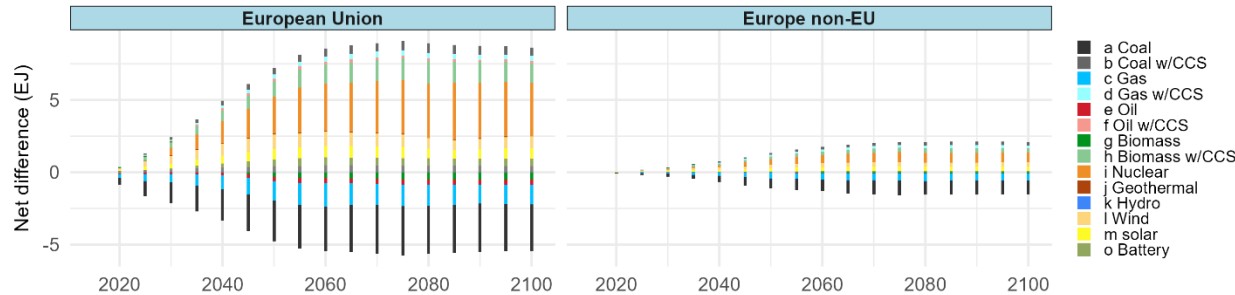

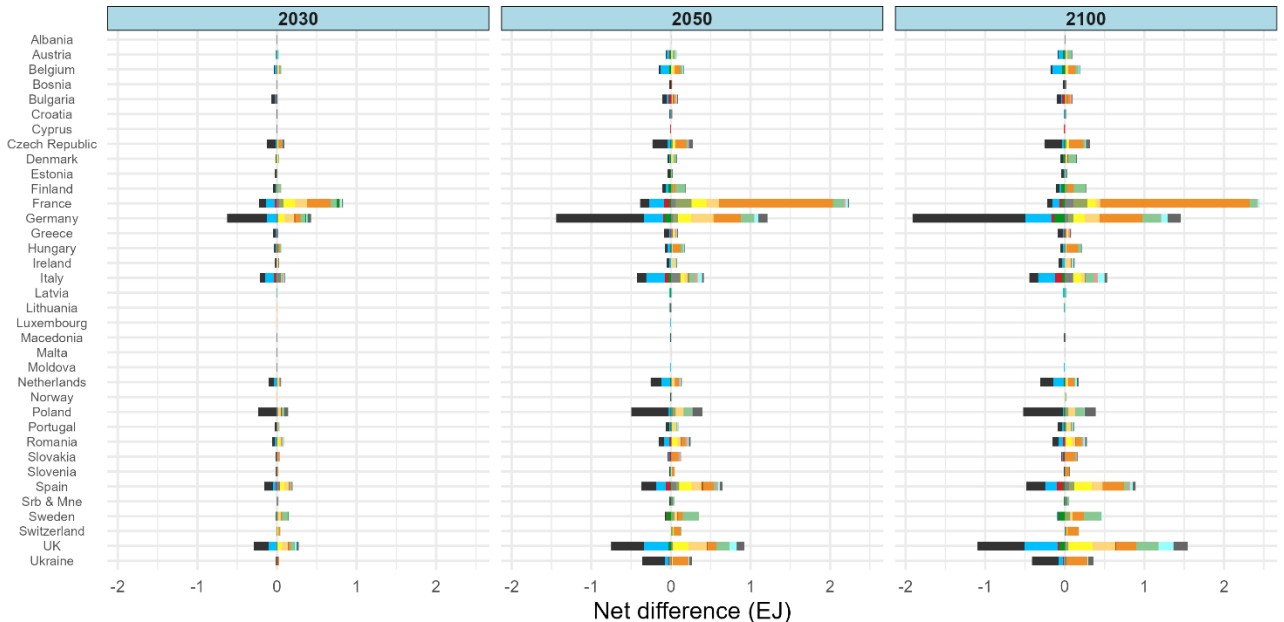

**Figure 14: Difference in electricity demand between the policy and the baseline scenarios by period, region, and technology.** The
upper panel shows absolute differences (*Policy - Baseline*) in electricity generation for the entire EU and non-EU regions by period, region
and technology (EJ). The lower panel shows the country level differences by period and technology (EJ). Belarus, Iceland and Turkey are
not included as they are not connected to the European grid regions.





Projected changes in the total energy demand and the composition of the energy mix will have direct implications for the emission of GHG and air pollutants (Figure 15 and Figure 16). CO2 emissions decline substantially by the end of the century, driven by reduced fossil fuel demand in both the EU bloc and the non-EU European region. Under the policy scenario, emissions in these regions decrease by 910 MTC and 362 MTC, respectively by 2100. This reduction is particularly

pronounced in several Scandinavian countries such as Denmark, Sweden, and Finland, where CO2 emissions fall by more than 250% by 2100. Significant reductions are also observed in other greenhouse gases (GHGs) as a result of the implementation of the policy. For example, methane emissions decrease by 7.0 Tg in the EU and 5.4 Tg in non-EU European regions by the end of the century. The most pronounced methane reductions are observed in Cyprus and Hungary, with decreases of over 60% by 2100.

Most air pollutants show a decreasing trend in the policy scenario compared to the baseline over the entire century. Nitrogen oxides (NOx), highly related to the combustion of the fossil fuels, particularly in the transportation sector, show a steady decrease at both EU and European non-EU region. By the end of the century, countries such as the Czech Republic and the UK reduce their $NO_x$ emissions by more than 50%. Similarly, $SO_2$ emissions, primarily associated with coal use, also decline across Europe over the entire time horizon. In some Eastern European countries, where coal dependency is high in the baseline

scenario, $SO_2$ emissions decrease by more than 70% by 2100 when the climate policy is implemented. However, the trends for other pollutants varies over time. The large-scale deployment of biomass technologies in the near term leads to an increase in certain biomass-related emissions, such as organic carbon (OC), by 2030. In some South-Eastern European countries, such as Bulgaria, Romania, or Italy, OC emissions rise by more than 50% by 2030. This could contribute to localized air pollution issues, highlighting the importance of integrating air quality and public health considerations into climate policy design

(Vandyck et al., 2021).



**Figure 15: Differences in GHG and air pollutant emissions between the policy and the baseline scenarios by period, region, and specie (MTC and Tg).**







**Figure 16: Percentage differences in GHG and air pollutant emissions between the policy and the baseline scenarios by period, country, and specie (%).**




## 5. Discussion and conclusion

GCAM-Europe represents a significant advancement for both academic and policy-focused communities. With its improved geographical and sectoral detail, alongside an integrated representation of energy, water, and emissions systems, GCAM-Europe offers a robust platform for comprehensive climate and environmental analysis. The model supports detailed assessments of climate policy impacts both between European countries and within them, reflecting regional differences and sector-specific complexities. Moreover, the model operates within the global version, allowing users to explore how European policies may influence, and be influenced by, developments in the rest of the world. GCAM-Europe's level of disaggregation, combined with its open-source framework, makes it especially valuable to a broad range of stakeholders, ranging from academic institutions to government agencies, NGOs, and private-sector experts. Particularly, the model is well-suited to support policymakers in the design and evaluation of alternative environmental and climate policies in a transparent and evidence-based manner.

This higher level of disaggregation also introduces certain challenges compared to the core version. The expanded number of markets significantly increases computational demands, particularly for those scenarios that incorporate climate policies, such carbon prices or exogenous emission caps. Likewise, the greater complexity introduced by the expanded set of regions, sectors and markets makes the model more sensitive and less robust to the implementation of new features, potentially leading to a higher likelihood of solution challenges compared to the core version.

Several improvements and new developments are planned for future versions of GCAM-Europe. The model is planned to remain under continuous development, progressively integrating features from successive releases of the core GCAM model. In the near term, the most immediate update will be the incorporation of features from GCAM v8.2 (JGCRI, 2025), which includes the shift of the final calibration year to 2021. Regarding consumer heterogeneity, the core GCAM model includes multiple consumers in the form of income deciles only in the residential sector, and allocates different residential energy services across consumers using exogenous assumptions based on the form of the demand function within each of the sectors. GCAM-Europe is planned to incorporate multiple consumers in all end-use sectors, namely food, transportation and municipal water. The within-region allocation of energy services across consumers will be improved by using country-level empirical data from the country-level Household Budget Surveys (HBS), which are available for all member states. In addition, we plan to explore new dimensions of consumer heterogeneity beyond income groups, including urban–rural distinctions, gender, and subregional classifications (e.g., NUTS levels). Finally, additional model expansion plans are under consideration, contingent on data availability. These include endogenizing industrial material use and building efficiency improvements, enhancing the representation of technology adoption, and refining consumer-specific elasticities to better capture behavioural responses. Further planned developments also involve sectoral and technological refinements, such as disaggregating diesel and gasoline vehicles, improving the representation of district heating, or incorporating hydrogen trade.



**Data and code availability:** The analysis has been developed using GCAM-Europe, which can be downloaded from the
following open-access online repository: https://github.com/bc3LC-GCAMEurope/gcam-core. The repository includes all the
equations, assumptions, and parameters that are read in by the model. The GCAM-Europe v7.2.0 release is available on
Zenodo: https://zenodo.org/records/15655568 (Sampedro et al., 2025).

A detailed documentation for all the input assumptions used in the core GCAM model can be found in the following open-
access repository: https://github.com/JGCRI/gcam-doc.

The code for reproducing the results and generating the figures has been stored in an open-access repository:
https://github.com/bc3LC-GCAMEurope/sampedro_etal_gcameurope

**Author contribution:** All the authors contributed to the development of the model. JS wrote the original draft and RH, CR
and DV contributed to Writing (review and editing). DV contributed to Funding acquisition.

**Competing interests:** The authors declare that they have no conflict of interest.

**Disclaimer.** The views and opinions expressed in this paper are those of the authors alone.

**Acknowledgements:** The authors acknowledge the entire GCAM development team at the Joint Global Change Research
Institute and particularly Pralit Patel, for their support during the development of the model. The authors also acknowledge
Natasha Frilingou, Konstantinos Koasidis, and Alexandros Nikas for their support in the model validation activities. The
authors acknowledge the use of Artificial Intelligence (AI), exclusively for language and grammar checks.

**Financial support:** This research is supported by the Horizon Europe European Commission Project 'DIAMOND' (grant no.
101081179). The authors also acknowledge financial support from María de Maeztu Excellence Unit 2023-2027 Ref.
CEX2021-001201-M, funded by MCIN/AEI /10.13039/501100011033; and by the Basque Government through the BERC
2022-2025 program. JS acknowledges financial support from the European Union's Horizon research program under grant
agreement 101060679 (GRAPHICS project).



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
