# Peer review of "GCAM-Europe v7.2.0: Enhancing Policy-Relevant Climate Modelling Through Spatial and Sectoral Detail"

_EGUsphere, 2025_

## Author Comment (AC1)

**Referee #1**

Source: https://egusphere.copernicus.org/preprints/2025/egusphere-2025-3546/#discussion

This article presents an enhanced regionalized version of GCAM, focused on Europe, and compares it to the standard GCAM regional representation. The reviewer acknowledges the substantial effort required to develop and document this contribution, both from a modelling and data perspective. Given that regional detail is crucial for accurately representing national decarbonization strategies, the topic addressed is scientifically relevant, and the improvements presented can significantly increase the scientific and political value of the GCAM Integrated Assessment Model applications.

Thank you for your thoughtful and detailed feedback on our manuscript. Your comments are greatly appreciated and have been very helpful in improving the clarity and quality of the study. Please find below our detailed responses to each of the points you have raised.

However, the comparison between the standard and extended geographic formulations is presented in a largely descriptive manner, with limited critical assessment. The manuscript does not sufficiently address the trade-offs between the two approaches. While results are reported at both European-aggregated and individual country levels, the core focus of the paper is on comparing geographic formulations. A more systematic comparison would therefore come from contrasting standard GCAM regional results with the aggregation of the Europe-detailed version. Several figures (e.g., Figures 4, 6, 7, 8, 9, 10, 11, 13, 14, and 16) would more strongly serve the paper's purpose if they showed comparisons at the five core European regions used in the standard GCAM formulation, rather than only country-level outcomes from the detailed model. Country-level results could be moved to supplementary material or omitted unless they demonstrate dynamics relevant for the comparison. The main text should then focus on differences across regions, highlighting benefits, trade-offs, or distinct dynamics revealed by the European-detailed formulation.

Thank you for this valuable comment. We agree that comparing the core GCAM version with GCAM-Europe is the central objective of these figures. While the comparison across the five European GCAM regions adds useful context, we consider the European-level comparison to be more illustrative (noting that we cannot separately distinguish EU vs. non-EU regions due to outdated mappings of Croatia and the UK in the core version). At the same time, we believe that country-level projections represent one of the key contributions of GCAM-Europe. For this reason,

we consider it important to retain these results in the main text. To reconcile these priorities and incorporate your recommendation, we have revised all figures using the following structure:

- **Main text figures**: Each figure now contains two panels. Panel (a) presents the comparison between GCAM-Europe and the GCAM core at the European level, while Panel (b) shows the country-level projections from GCAM-Europe. The two panels are clearly distinguished with explicit labels.
- **Supplementary Information**: For each result, we provide an additional figure showing the comparison between GCAM-Europe and the GCAM core for the five European regions.

We believe this approach strengthens the comparative analysis while preserving the country-level projections, which we view as essential outputs of the model. An illustrative example of the structure of the new figures can be found at the end of this document. We also highlight we have expanded the discussion of the results, incorporating (though not limited to) the suggestions raised in the "Results discussion" section of your review.

The authors briefly note only in the conclusion section that computational demands increase, that the model is more sensitive, and that it is less robust to the addition of new features. These points deserve fuller discussion in the main text, ideally in a dedicated section rather than only in the conclusions. For instance: What does "more sensitive" mean in practice (e.g., instability, difficulty converging)? In what ways is robustness reduced? How were these issues identified, and how might they be addressed?

Thank you for your comment. We agree that describing the technical requirements is a key part of the model documentation, as it directly affects usability. Therefore, we have added dedicated paragraphs under the "Model Description" section, detailing the model's computational needs and resources. In short, GCAM-Europe requires higher data processing capacity, which lowers efficiency and increases simulation time. The model also demands substantially more computational resources when addressing complex scenarios, such as those involving detailed climate policies. Consequently, some scenario runs take longer to complete. Previously, we referred to this as *computational sensitivity*, though we acknowledge that the term was imprecise and have therefore removed it.

"With its higher regional disaggregation (from 32 to 66 regions) and the additional layers of complexity (e.g., electricity grid regions or expanded household technologies), GCAM-Europe requires substantially more input data for setup and a larger number of markets and equations to solve at each time step. This added complexity makes the model considerably more memory intensive: it consumes roughly 50% more RAM than the core version and increases system-wide committed memory. As a result, GCAM-Europe requires the page file, which reduces CPU efficiency and slows computation. In practice, a full simulation run (e.g., the Reference scenario to 2100) takes about three times longer to complete. To run GCAM-Europe efficiently, systems with at least 128 GB of RAM and high memory

bandwidth are advisable, together with SSD (NVMe) storage to reduce paging overhead, since memory capacity and bandwidth remain the critical constraints."

The paper would benefit from a deeper discussion of the practical implications of higher geographic detail. Specifically:

Thank you for your comment. We have answered in detail to the specific questions below.

• How does the dimensionality of the problem change?

The increase in dimensionality operates on two fronts. First, the geographical resolution expands from 32 to 66 regions, multiplying the number of markets, flows, and equilibrium equations that must be solved at each time step. Second, the introduction of additional household categories, electricity grid regions, and technology vintages further enlarges the model's resolution space, adding complexity across multiple dimensions.

• Does the model require significantly more time or memory to solve?

GCAM-Europe 7.2 requires approximately 50% more RAM than GCAM-Core 7.2, pushing the system closer to memory limits and increasing reliance on the paging file. This reduces CPU efficiency and makes simulations considerably slower; for instance, a Reference run to 2100 takes roughly three times longer to complete compared to a Reference run in GCAM-Core.

• Is there a trade-off between adding geographic detail and maintaining detail in other dimensions (e.g., technology, resources)?

The design philosophy of GCAM-Europe has never been to simplify existing dimensions in order to accommodate new ones. Instead, geographic disaggregation and technological advancements have been developed on top of the GCAM-Core structure. This approach preserves the full detail of GCAM-Core but results in greater technical demands and longer runtimes, as no simplifications of the core framework have been introduced.

 Would this be a step towards replacing the standard GCAM formulation with a higher geographical detail version or the benefits for other regions are not sufficient to justify such change? Which are the steps or processes necessary to achieve such goal?

The objective of GCAM-Europe is not to replace the standard global GCAM formulation, but rather to provide a robust regionalized version of the model that can better address Europe-specific research questions. Adding this level of geographical detail to the core global model would introduce unnecessary complexity for a model designed to maintain a broad, global perspective. Instead, GCAM-Europe is part of the suite of regional extensions (such as GCAM-USA and GCAM-

China) that follow a consistent structure and can be run independently. Together, these versions complement the core global model by enabling detailed analyses at regional scales while preserving the tractability and global scope of the standard formulation.

We note that, in the revised version of the manuscript, we have improved the Discussion section, expanding the technical discussion by incorporating additional details and clarifications directly informed by the issues raised in this comment:

"This higher level of disaggregation aims to better address Europe-specific research questions, but it does not aim to substitute the current representation of the European continent in the GCAM-Core version. The extra level of complexity introduces additional technical challenges that are unnecessary for a model designed to maintain a global perspective to have a broad global perspective. In particular, the expanded number of markets and the new electricity grid structure make the model significantly more memory-intensive, resulting in longer overall run times. As in other models, the inclusion of policy scenarios further constrains the system, increasing computational demands. In the case of GCAM-Europe, these demands can reach technical thresholds that make it more suitable to run in a cluster computer."

Additionally, it would be valuable to clarify which challenges are specific to the EU disaggregation exercise and which can be generalized to other GCAM or IAM regional disaggregation efforts. Lessons learned and recommendations for future work would further increase the impact of the study.

Thanks for raising this point. In the revised discussion section, we now explicitly distinguish between challenges that are largely data-related and therefore relevant to other IAM and GCAM regional disaggregation efforts, and those that are specific to the GCAM framework, as noted in earlier responses.

"Beyond computational challenges, the development of a regional version of the model also entails important datarelated difficulties that are likely to be relevant for other modelling efforts. In particular, the absence of country-level information in key databases (e.g., Eurostat) often necessitates filling data gaps with alternative sources, such as global balances or national statistics. This process requires extensive harmonization to ensure consistency across datasets and represents a substantial component of the overall modelling effort."

**Additional comments:**

The introduction could better situate this work in the broader IAM literature, including references to other models' approaches to geographic detail and to applications in IPCC assessments.

Thank you for the excellent suggestion; it has greatly assisted us in improving the positioning of our contribution and better flesh out the novelty of GCAM-Europe. The introduction has been

heavily revised, while in the Supplementary Material we provided a table (Table S1) with a comparison of the regional granularity, methodological approach and openness of the models with at least one scenario in the IPCC AR6. We first improved the discussion on the role of IAMs in recent IPCC reports that has drastically increased:

"Notably, from the hundreds of scenarios included in the early IPCC reports (van Beek et al., 2020), the contribution of Working Group III in the recent Six Assessment Report (AR6) of IPCC featured a little more than 1,200 scenarios, and almost thrice as many submissions (Kikstra et al., 2022)."

We then expand the discussion on openness, notably mentioning that within the models included in IPCC AR6, GCAM is among the very few that are fully open source and at the same time accessible. Although the community has made improvements, many models still rely on proprietary software (e.g., optimization software like GAMS), a limitation that does not apply in the case of GCAM, and consequently GCAM-Europe.

"A key advantage of GCAM is that it is a fully open-source and accessible model supported by an active and widespread community of practice. Although the IAM community has recently made progress in releasing the source code of many emblematic models, the reliance on proprietary support software remains a core limitation, particularly in the case of optimisation models. By adopting a price-clearing, recursive-dynamic, partial-equilibrium solution mechanism, GCAM has positioned itself among the few models that can be freely used by any potential user without the need for proprietary software (see Table SX). This openness has fostered a strong community that ensures continuous maintenance, updates, and expansions."

We then proceed to further add arguments on the limited spatial representation on the EU, with references to specific models that address this gap, and their accompanying limitations (e.g., closed source and/or relying on proprietary data).

"Despite their usefulness in climate science, IAMs have also faced significant criticisms (Ackerman et al., 2009). One common criticism is on the spatial granularity represented"

"In fact, out of the 13 models with at least one scenario in IPCC AR6 (Sognnaes and Peters, 2025), only two global models featured a disaggregated representation of all EU member states (see Table SX), namely POLES (Criqui et al., 2015), a closed-source energy system model, and GEM-E3 (Capros et al., 2013) a general equilibrium model, relying on the proprietary Global Trade Analysis Project (GTAP) database (Aguilar et al., 2022). On the other hand, regional bottom-up models with a detailed representation of the EU, like EU-TIMES (JRC, 2013), often lack the ability to capture global trends along with the regional dynamics".

Based on the above, we finally explicitly state the gap that GCAM-Europe aims to address, while improving the introduction of our contribution in the form of delivering the model.

"However, neither the GCAM nor the broader IAM community has so far managed to deliver a European version of a core global model, with full member state granularity that is open source and freely accessible, despite the EU's historical international climate leadership (Oberthür and Dupont, 2021)."

"Within this context, we present GCAM-Europe a powerful tool for in-depth analysis of European climate policy at the member state level which, unlike existing regional versions of GCAM, explicitly represents land use, agriculture, water resources, and energy extraction at the subregional level. This constitutes a significant contribution to the European IAM community, enabling more precise assessments of resource dynamics and policy impacts across diverse European contexts, with the potential to receive strong interest from key stakeholders involved in the climate transition of the EU and globally".

**Comments on trade:**

- Further clarification is needed regarding the multilevel Armington trade formulation:
- How are elasticities of trade determined?
- Are there differences between EU and member state parameterizations and methods?
- How do the two Armington levels interact?
- Was any parameter adaptation required for consistency across levels?
- Are there published literature describing this formulation?

Thank you for these comments, we have expanded the section explaining the Armington trade formulation, referring to literature describing the default GCAM estimation of such elasticities, and how parametrizations have been adapted for intra-EEA trade. The double Armington structure has not been published as such, but defined on logical modifications on the pre-existing structure, overall assuming intra-EEA trade to occur with less frictions than extra-EEA trade for EEA countries.

"By default, commodity trade in GCAM, including energy, food, and selected industrial commodities such as steel and ammonia, is governed by Armington trade specifications (Armington, 1969; Zhao et al., 2022), which define the substitutability between domestically produced and imported goods, both at the international level and within individual regions. Armington trade elasticities are commodity-specific and determined by literature (Hertel et al., 2007) and hindcasting experiments (Zhao et al., 2021). In GCAM-Europe, EEA countries are treated as a single trading block in relation to other GCAM regions in the international market, including the newly separated United Kingdom. Within the EEA, intra-regional trade is modelled with an additional layer of Armington specifications to capture trade flows among member countries. While international trade costs and Armington elasticities between the EEA block and non-EEA regions remain the same as in the default GCAM framework, trade within the EEA is assigned lower costs and friction (e.g. higher elasticities), reflecting reduced trade barriers within the internal markets. For those commodities where trade costs apply, a rule of thumb is applied, determining intra-EEA trade (e.g. between France and Germany) to cost 1/3rd of extra-EEA (e.g. between France and USA), reflecting closer distances and lower administrative costs. In terms of trade elasticity, intra-EEA crop trade is assumed to be frictionless, having EEA-wide crop commodity markets rather than national markets. Intra-EEA trade for other commodities (animal, energy and industrial commodities) is assumed to occur with very low friction (logit of -12). Beyond improved trade realism, this enhanced structure facilitates the representation of shared trade policies across the EU and EEA, such as the Carbon Border Adjustment Mechanism (CBAM), enabling more nuanced and policy-relevant modelling in future analyses."

The authors should clarify whether the improved building-sector coverage could (or should) be integrated into the core GCAM formulation, and whether data availability is a limiting factor for that.

Thank you for the comment. This has been clarified in the revised version of the manuscript. Note that the new text has been included at the end of Section 2 and not at the section on building coverage (2.3), as we believe that the point raised by the reviewer is extendible to other sectors in the model.

"Although limited data availability at the global scale constrains the level of sectoral detail in the core model, particularly in areas such as buildings, greater regional data availability allows for more granular sectoral representation in regional model versions like GCAM-Europe and GCAM-USA."

Regarding consumer heterogeneity in the residential energy sector, could the authors cite the data sources and assumptions underlying the use of income deciles? Since Household Budget Surveys (HBS) are only proposed for future work, mentioning it in the buildings results analysis serves little purpose for the paper discussion in this reviewer opinion and it would be more appropriate to be part only of the conclusions section as a suggestion for extensions rather than in the main results.

Thank you for this valuable comment. We agree that the near-term incorporation of Household Budget Surveys (HBS) and their potential benefits are more appropriately discussed in the Discussion section, rather than in the main model description. In response, we have revised Section 2.3 to provide a clearer description of the current implementation of consumer heterogeneity in the residential sector, including the underlying assumptions underlying. The mention of HBS has been streamlined to a brief forward-looking note, with the more detailed discussion of its role and potential implications moved to the Discussion section:

"In terms of household structure, GCAM-Europe includes consumer heterogeneity in the residential energy sector in the form of income deciles, which is aligned with the current structure in the core version of the model. To implement the multiple consumer groups, we adopt the core GCAM approach by applying the corresponding demand functions, calibrated to average income data, to estimate consumption levels for each income decile within a region. Because the aggregation of decile-level estimates does not necessarily match with the observed national (or regional) consumption, a bias-correction term is introduced, defined as the difference between the estimated aggregate and the observed value, he details of this implementation in core GCAM are documented in Sampedro et al. (2024). As a next step, we plan to enhance this representation by integrating empirical data from national Household Budget Surveys (HBS) (Eurostat, 2025c), which are available for all EU member states (see Discussion)."

For socioeconomics, it would be helpful to briefly note challenges of extending this formulation beyond SSP2.

We share the reviewer's belief that choosing to adapt to different SSPs might be more challenging than just plugging in different socioeconomic projections. We view two main challenges. First, the EU Ageing Report 2024 provides only one baseline demographic projection, not a range of alternative scenarios. Therefore, our current population projections are consistent with SSP2 but also incorporate EU-specific details from the Ageing Report. Switching to another SSP would likely require abandoning this EU-specific adaptation and reverting to the native SSP population data at the national level. Second, fully aligning with a different SSP storyline would require not just replacing GDP and population input, but techno-economic parameters and scenario drivers, as well as any other elements to harmonise to the underlying narrative of each SSP. We reflect on this in the following addition:

"In principle, it is possible to adapt socioeconomic assumptions to other SSPs. However, the EU Ageing Report 2024 provides only one baseline demographic projection that is generally more consistent with SSP2, and not a range of alternative scenarios like the SSPs. Opting for different SSP socio-economic assumptions, might entail avoiding the adaptation to EU-specific data, and reverting to the native SSP population data at the national level. Additionally, fully harmonising a model to different SSP storylines would require not just adapting socio-economic but also technoeconomic assumptions and scenario drivers both at the global level (core) and the EU part".

Some sentences lack context, such as: "This reduction is particularly pronounced in several Scandinavian countries such as Denmark, Sweden, and Finland, where CO2 emissions fall by more than 250% by 2100" Relative to which year or scenario? Absolute values or clearer framing would improve readability.

Thank you for this comment. We agree that the original sentence lacked sufficient context. In this section, our intention was to show relative reductions in CO2 emissions, as we consider them more informative than absolute values when comparing country-level mitigation efforts. However, we recognize that clearer framing was needed. We have therefore revised the sentence and have adjusted the wording to improve readability.

"This reduction is particularly pronounced in several Scandinavian countries such as Denmark, Sweden, and Finland, where CO2 emissions in the policy scenario fall by more than 250% by 2100, compared to the baseline."

In addition, we have carefully reviewed the entire section and made further revisions to ensure that the presentation of results is consistently framed.

The authors mention moving the calibration year to 2021 as future work. What calibration year is used in the present paper?

The revised manuscript clarifies that the final calibration year in the current GCAM-Europe version is 2015 and discusses how updating this parameter improves future model projections:

"In the near term, the most immediate update will be the incorporation of features from GCAM v8.2 (JGCRI, 2025), which includes the shift of the final calibration year to 2021. Since the current version of GCAM-Europe is calibrated to 2015, adopting the updated calibration year is expected to enhance the reliability of future projections."

**Results discussion**

Thank you for these comments. We have answered in detail to the specific questions below.

**Biomass**: Since biomass use increases, the paper should clarify how much is domestically sourced versus imported when it is first mentioned. Europe's limited biomass potential is documented in other literature (e.g., Ruiz et al. 2019, ENSPRESO database, <a href="https://doi.org/10.1016/j.esr.2019.100379">https://doi.org/10.1016/j.esr.2019.100379</a>), and results should be contextualized against such benchmarks.

The updated trade figure presents biomass consumption, domestic production, and trade flows at the European level, along with a more detailed analysis by country. Similar to other commodities, the figure compares results from GCAM-Europe with those from the default GCAM core version (add results). Following the reviewers' recommendation, we also compare the estimated biomass production levels with the values reported in Ruiz et al. (2019). Our results indicate that the projected biomass deployment (approximately 17 EJ) is between the reference and the "high potential" scenarios presented in Ruiz et al. (2019).

"Biomass production rises in GCAM-Europe relative to the core model (17 EJ versus 15 EJ), consistent with recent estimates of European biomass potential (Ruiz et al., 2019). Although intra-EEA biomass trade exists, it remains limited compared to international exchanges. At the country level, Scandinavian nations are major biomass exporters, with Sweden exporting about 1.6 EJ in 2050".

**Industry**: Is industrial leakage to other regions considered? Is there any change between the formulations?

As in the core version of GCAM, industrial leakage can occur for commodities where trade is explicitly represented, like iron and steel, and ammonia. The probability for industrial leakage

increases with higher price differences between domestic markets. The updated trade figure shows however that, relative to the GCAM-core version, a larger share of industrial commodity trade in GCAM-Europe occurs between EEA countries, reducing trade/leakage with/to non-EEA countries. This behavior is caused by the assumed low trade frictions within EEA countries, motivating potential leakage/relocation to occur within the EEA.

**Electricity**: Why does wind's share decrease so substantially by 2050? Which technologies replace it, and why? Is this due to costs, learning-by-doing, or artifacts of regional disaggregation?

The low share is mainly driven by the relatively low uptake in the base year (2015) relative to its costs in that year, which were already low and are assumed to only marginally decrease up to 2050. The low update feeds through future periods as a low "stated preference" for wind. Nevertheless, when climate policy is applied, its share does increase significantly due to relative tax benefits.

**Land use**: What drives changes in grazed vs. non-grazed pastures, and in managed vs. unmanaged forests? Are these cost-driven or flat solution space artefacts of the land-use representation?

Basically, separating out countries reduces the potential bias in these shares. We added some sentences in chapter 3.3 to describe this logic:

"These differences are driven by the use of assumed yields in combination with observed national outputs of forest and animal products to define the relative share of managed pasture and forest relative to all pasture and forest (Zhao & Wise, 2023). Real-world deviations from the assumed yields can lead to under- or overestimation of "managed land", and by aggregating many regions, such under or overestimations can persist throughout the whole region. Having separated all countries however, the estimated area of "managed land" on aggregate are likely more accurate, as the impact of potential yield deviations are limited to each individual country."

**Emissions**: presenting absolute values of country emissions in the new formulation provides limited insight. Relative emission shares and changes across regions between the two model formulations would be more informative for assessing how burden-sharing dynamics shift under increased geographical detail.

We have revised the presentation of the emissions results in line with your suggestion and for consistency with other sections of the manuscript. In the main text, the updated figure now includes two clearly differentiated panels: (a) differences in emissions between GCAM-Europe and

the core at the European level, and (b) a snapshot of emissions of different species in 2050 for individual countries in GCAM-Europe. In addition, and following your recommendation, we have added a new figure (Figure S14 in the SI) that shows relative changes across regions between GCAM-Europe and the core for the original European regions in the core model.

**Trade effects**: Since the authors state that the multi-level Armington formulation influences import patterns for goods like ammonia, iron, steel, and so on; it would be important to also present in the paper marginal costs (or prices) and the imports levels versus internal production, so we could confirm if the change in traded amounts is substantial or not to allow the comparison between both scenarios results. Is trade is responsible to too much change in the energy sector, the comparison between both scenarios provides less insight, and the changes are mainly driven by the different Armington trade formulation, instead of the increased internal detail of the energy system.

The updated trade figure now provides greater detail on domestic production as well as intra-EEA and extra-EEA trade, enabling a clearer comparison between the two model configurations. While we appreciate the suggestion, we believe that incorporating marginal costs for these commodities would not substantially change the interpretation of the results, and have therefore chosen not to include them. Prices in GCAM are calibrated, and only relative price differences into the future (e.g. driven by input costs and/or policies) are driving trade levels over time. The results show that the adapted trade structure leads to less extra-EEA trade overall, driven by the new Armington formulation reflecting the relative benefits of keeping trade within EEA borders.

**Results realism**: The policy scenario results attribute increased nuclear expansion to Germany and France. Is this politically realistic given Germany's nuclear phase-out? Were political or legislative constraints considered for nuclear and other technologies?

This is an interesting observation, and the reviewer is correct that nuclear energy in Germany increasing goes against the phase out efforts based on the Energiewende. We argue that although this may render the scenario less realistic (at least in this specific dimension) this is still an important insight for the model itself, especially given that the policy scenario represents a stylized EU-wide target. First, when introducing a new model, it is important to highlight all the inherent tendencies of the tool, to inform future users of potential assumptions that need to be taken. Second, this highlights a critical gap in existing modeling efforts: aggregate EU results from models without member state—level disaggregation might appear reasonable at the EU level, while concealing underlying regional outcomes that would be politically unrealistic. Third, it underscores the potential future uses of GCAM-Europe, particularly for exploring national policies at the

member-state level, which would yield more policy-realistic results. We reflect on this in the following addition:

"In the case of Germany, this result seems counterintuitive, given the country's long-standing effort to phase-out nuclear energy as part of the Energiewende strategy (Rogge and Johnstone, 2017). However, the policy scenario implemented here is fairly stylized in the sense that it applies an EU-wide target, with the model then freely making decisions based on a cost-optimal basis. This is an important insight both in terms of transparency and prospect capabilities of the model. First, even a version of the model without member state—level spatial granularity might yield similar aggregate EU results, while concealing an underlying regional allocation that would, in practice, be politically unrealistic. By transparently providing the unconstrained tendencies and model fingerprints (Dekker et al., 2023) of the newly developed model, future users are informed on appropriate assumptions needed and consequently develop more realistic scenarios. Second, it also highlights the importance of region-specific modelling, since applying national-based constraints and targets per country would improve the policy-realism of the scenarios compared to the aggregated cost-optimal ones."

**a) Comparison between GCAM versions**

**b) Country-level projections in GCAM-Europe**

**Figure 1: Domestic production and trade dynamics in 2050 by scenario and commodity (EJ or Mt).** Panel A shows total emissions for the European continent by scenario, period and specie (MTC and Tg). Panel B shows emissions by period, country and specie for 2050 in the GCAM-Europe baseline. A comparison between GCAM-Europe and the GCAM core for the five core European regions is provided in Fig S15. Panel B does not disaggregate intra and extra EEA trade.

Figure S2: Domestic production and trade dynamics in 2050 by scenario, commodity and core European region.